# Conformational Remodeling and Allosteric Regulation Underlying EGFR Mutant-Induced Activation: A Multi-Scale Analysis Using MD, MSMs, and NRI

**DOI:** 10.3390/ijms26136226

**Published:** 2025-06-27

**Authors:** Hui Duan, De-Rui Zhao, Meng-Ting Liu, Li-Quan Yang, Peng Sang

**Affiliations:** 1College of Agriculture and Biological Science, Dali University, Dali 671000, China; dhui4399@gmail.com (H.D.); zderay945@gmail.com (D.-R.Z.); lminimt@163.com (M.-T.L.); 2Key Laboratory of Bioinformatics and Computational Biology of the Department of Education of Yunnan Province, Dali University, Dali 671000, China; 3Co-Innovation Center for Cangshan Mountain and Erhai Lake Integrated Protection and Green Development of Yunnan Province, Dali University, Dali 671000, China

**Keywords:** EGFR, conformational dynamics, Markov state models, neural relational inference, allosteric regulation

## Abstract

Activating mutations in the epidermal growth factor receptor (EGFR) are key oncogenic drivers across multiple cancers, yet the structural mechanisms by which these mutations promote persistent receptor activation remain elusive. Here, we investigate how three clinically relevant mutations—T790M, L858R, and the T790M_L858R double mutant—reshape EGFR’s conformational ensemble and regulatory network architecture. Using multiscale molecular simulations and kinetic modeling, we show that these mutations, particularly in combination, enhance flexibility in the αC-helix and A-loop, favoring activation-competent states. Markov state modeling reveals a shift in equilibrium toward active macrostates and accelerated transitions between metastable conformations. To resolve the underlying coordination mechanism, we apply neural relational inference to reconstruct time-dependent interaction networks, uncovering the mutation-induced rewiring of allosteric pathways linking distant regulatory regions. This coupling of conformational redistribution with network remodeling provides a mechanistic rationale for sustained EGFR activation and suggests new opportunities for targeting dynamically organized allosteric circuits in therapeutic design.

## 1. Introduction

The epidermal growth factor receptor (EGFR) is a prototypical member of the receptor tyrosine kinase family and plays a central role in regulating cell proliferation, survival, differentiation, and migration [1]. Its activation is initiated by ligand binding, which induces receptor dimerization and triggers intracellular kinase activity. Mutations in the EGFR gene, particularly those occurring in the kinase domain, are frequently observed in various malignancies—including non-small cell lung cancer (NSCLC) and glioblastoma [2,3], where they often act as key oncogenic drivers. These mutations not only promote tumorigenesis but also profoundly influence cellular sensitivity to chemotherapeutic agents and targeted therapies [4]. Understanding the molecular mechanisms by which EGFR is activated, and how specific mutations modulate this activation, remains a fundamental goal in cancer biology with direct implications for precision oncology.

Structurally, EGFR comprises three major components: an extracellular domain that binds epidermal growth factor (EGF), a transmembrane helix anchoring the receptor in the cell membrane, and an intracellular catalytic domain (CD) responsible for signal transduction through tyrosine autophosphorylation. Upon EGF binding, EGFR undergoes activation, involving the formation of an asymmetric dimer in which the C-lobe of one kinase domain interacts with the N-lobe of the adjacent domain, stabilizing the active conformation. This transition is orchestrated by coordinated movements within several conserved structural elements, including the activation loop (A-loop), the αC-helix, the phosphate-binding loop (P-loop), and the DFG motif.

The A-loop, located in the N-lobe, plays a pivotal role in kinase activation. In the inactive state, the A-loop adopts a folded conformation, with a short α-helix at its N-terminus that interacts with the αC-helix, thereby inhibiting activation. The αC-helix in the inactive state is rotated outward (αC-out) [5], preventing key stabilizing interactions. Upon activation, the A-loop extends, forming a hairpin-like structure, while the αC-helix shifts inward (αC-in), and a salt bridge forms between Lys745 and Glu762, stabilizing the active configuration. Additionally, the P-loop, located in the N-lobe, undergoes a conformational change that facilitates ATP binding, crucial for EGFR’s catalytic activity. Another essential structural element is the DFG motif (Asp-Phe-Gly), which plays a central role in regulating ATP binding [6]. In the inactive state, the DFG motif adopts a “DFG-out” [7] conformation that prevents ATP binding, while in the active state, it shifts inward to the “DFG-in” conformation, allowing ATP to bind and activating the kinase. These coordinated changes in the A-loop, αC-helix, P-loop, and DFG motif enable EGFR to transition from an inactive to an active state, a process frequently disrupted by oncogenic mutations that favor activation-prone conformations and destabilize the inactive state. Figure 1 illustrates these dynamic transformations and highlights the roles of these key structural components during EGFR activation.

Several recurrent somatic mutations within the EGFR kinase domain, most notably L858R and T790M, have been shown to significantly alter the conformational landscape of the receptor. The L858R mutation, located within the activation loop, enhances catalytic activity by stabilizing the active conformation and promoting receptor dimerization [8]. Mechanistically, it is thought to favor the extended conformation of the A-loop and prevent residue 858 and its surrounding region from adopting the α-helical structure associated with the inactive state. Long-timescale molecular dynamics (MD) simulations further suggest that L858R may reduce the intrinsic disorder of the αC-helix in the receiver N-lobe, thereby indirectly promoting activation [9].

The T790M mutation, situated at the gatekeeper position in the ATP-binding pocket, is a canonical example of acquired resistance to tyrosine kinase inhibitors (TKIs) [10]. Initially thought to interfere with drug binding via steric hindrance, it is now understood to confer resistance primarily by increasing ATP affinity, which diminishes inhibitor effectiveness. When combined with L858R, the double mutant exhibits greater ATP binding capacity and resistance than either single mutant alone, suggesting a synergistic effect that further promotes an activation-prone state.

While the structural consequences of these mutations have been widely studied, most existing work has focused on static conformational preferences or the reshaping of the free-energy landscape [11]. Recent studies have proposed that activating and resistance mutations may stabilize active-like states through local structural rearrangements and interdomain interaction [12]. However, a comprehensive understanding of how such mutations promote activation—particularly through changes in thermodynamic stability, conformational kinetics, and allosteric signaling remains elusive. The activation process of EGFR is not a simple binary switch; rather, it involves a complex series of conformational changes that occur along a continuum of intermediate states [13]. These transient intermediates are often sparsely populated and difficult to capture using conventional structural techniques, making it challenging to fully understand the activation mechanism at an atomic level. Moreover, the role of mutations in shifting the equilibrium toward active states and accelerating conformational transitions remains poorly characterized.

This study aims to elucidate how activating mutations alter the conformational dynamics of EGFR and modulate its activation. By integrating MD simulations, metadynamics simulations, Markov state models (MSMs), and neural relational inference (NRI), we provide new insights into how these mutations influence EGFR’s conformational ensemble, favoring activation-prone states and facilitating transition to the active conformation. Furthermore, we investigate the reorganization of the allosteric network that accompanies these mutations, shedding light on the role of long-range domain coupling in driving EGFR activation.

## 2. Results

### 2.1. Structural Flexibility and Molecular Motion Analysis

To assess the global structural stability of EGFR variants, we calculated the root-mean-square deviation (RMSD) of backbone atoms relative to the initial structure. WT-EGFR quickly stabilized and maintained low RMSD values, whereas T790M and T790M_L858R mutants exhibited larger deviations and slower convergence, indicating mutation-induced destabilization (Appendix A). To characterize residue-level flexibility, then we computed the root-mean-square fluctuation (RMSF) of Cα atoms across all systems (Figure 2). Although the overall RMSF averages were slightly lower for mutants, local fluctuations revealed notable increases in functionally important regions. Figure 2a shows a comparison between T790M and WT. It is clear that the mutant has greater conformational flexibility in the αC-helix region than the wild type. Both L858R and T790M_L858R mutants exhibited elevated flexibility in the activation loop (A-loop, residues 845–860) and αC-helix (residues 760–770), which are essential regulators of EGFR activation. RMSF difference plots (Figure 2b,c) further highlighted mutation-induced flexibility gain in these regions relative to WT, suggesting that the mutations disrupt local constraints and increase structural adaptability. This enhanced local motion may facilitate repositioning of the αC-helix and unfolding of the A-loop, which are required for the transition to the active conformation. Figure 2d shows an overall comparison of RMSF values between the wild type and all mutant types.

To investigate collective conformational dynamics, we performed a principal component analysis (PCA) on the trajectory ensembles. Eigenvalue spectra (Appendix A) revealed that T790M-EGFR dynamics were concentrated in the first few principal components, with PC1–PC3 accounting for over 85% of the total variance. This indicates a more directed and coherent conformational trajectory compared to WT or L858R, which exhibited more dispersed variance profiles. Porcupine plots projected along PC1 were used to visualize the dominant motion vectors across functional regions (Figure 3). WT-EGFR displayed asynchronous and low-amplitude movements in the A-loop and P-loop, consistent with an inactive configuration. The L858R mutant showed a similar pattern, suggesting limited dynamic impact. In contrast, T790M and T790M_L858R mutants exhibited larger and partially aligned displacement vectors in the A-loop, P-loop, and αC-helix, implying enhanced coupling among regulatory elements. The T790M-induced vector field was oriented toward inward rotation of the αC-helix and extension of the A-loop—hallmarks of an activation-prone state. In the double mutant, displacement magnitude was preserved, but the directional distribution was altered, indicating that L858R may reshape the transition pathway initiated by T790M.

Together, these analyses demonstrate that activating mutations reconfigure the conformational flexibility and coordinated motion of EGFR, particularly by enhancing local mobility and long-range domain coupling, thereby promoting accessibility to active-like states.

### 2.2. Thermodynamics and Kinetics of Conformational Transitions

To delve into the thermodynamic and kinetic properties of EGFR conformational transitions, we constructed Markov state models (MSMs) for the MD trajectories of wild-type and mutant EGFR, respectively. We first used the VAMP-2 scoring method to determine the appropriate number of microstate clusters. By analyzing different numbers of clusters, we found that the trans-state equilibrium was optimal when the number of conformational clusters was 1000 microstates (Appendix A). This result provides a reasonable division of microstates for subsequent analyses and helps us to better reveal the transition process of EGFR from the inactive to the active state. We chose the implied time scale to guide the determination of the MSMs lag time, which was ultimately determined to be 2 ns, which maximizes the likelihood of capturing detailed information about the system’s slow motion while ensuring Markovianity (Appendix A).

While structural flexibility and molecular motion analyses suggest that the EGFR mutant shifts to a more open and active conformation, it remains uncertain whether the EGFR mutant simulation ultimately samples the active state structure. The active state EGFR structure (PDB ID: 2GS2) was used as a reference for the active conformation. As described in the introduction, the main difference between the active and inactive states of EGFR is the conformational difference between the A-loop region and the αC-helix region, which are responsible for open and closed conformations. Therefore, we chose the Cα-Cα distance between residues in the A-loop and αC-helix regions as an analytical feature.

To further elucidate the conformational dynamics of EGFR and the effects of mutation, we employed Markov state models (MSMs) combined with the use of PCCA+ (Perron Cluster Cluster Analysis) to group the microstates of the four MD simulated systems into four macrostates, as shown in Figure 4a–d. This approach allowed us to identify key substates and their transition pathways, leading to a deeper understanding of the thermodynamic and kinetic properties of wild-type and mutant EGFR. By using PCCA+, we categorized the microstates into four macrostates (S1–S4) based on their conformational similarities and jump probabilities. These macrostates represent the set of different conformations that capture the basic kinetics of EGFR. Where the macrostates are color-coded and mapped onto the first two independent components (IC1 and IC2) derived from a time lag independent component analysis (TICA). This visualization highlights the separation and distribution of macro-states in a simplified conformational space.

As shown in Figure 5, the equilibrium distributions of wild-type and mutant EGFR conformational states differ significantly. For example, the representative structure of state 4 in the WT-EGFR state is very similar to the initial structure of the simulated inactive conformational state and dominates all conformational groups, with a conformational ratio of up to 99.8%, whereas states 1, 2, and 3, which converge to the active state, collectively account for less than 1% (Figure 5a), suggesting that the wild-type state of the EGFR basically remains inactive throughout the simulated conformational state. As shown in Figure 5a, the transition times from states 1, 2, and 3 to state 4 are all very short, that is, the three conformations converging to the active state can be easily converted to the inactive state 4, whereas the corresponding reverse transition time (state 4 to the other three states) is significantly longer than the other three states. The corresponding reverse transition (state 4 to the other three states) time is significantly slower, in other words, it is difficult for WT-EGFR to convert to the active state during the simulation.

In the T790M-EGFR state, the representative structure of state 4 is partially similar to the initial structure of the active conformation, and the α-helix at the end of the A-loop appears to be partially de-helicalized, which is in the intermediate state of the transition from the inactive state to the active state, and the proportion of this conformation occupies 99.1% (Figure 5b), and the transition times from states 1, 2, and 3 to state 4 are 5.22 ns, 2.94 ns, and 3.08 ns, respectively, indicating that these three inactive conformations (states 1, 2, and 3) can easily transition to the active conformation (state 4) with a fast transition rate. This implies that these inactive states tend to converge to the active state quickly. In contrast, the inverse transition times from the active conformation (state 4) to the other inactive conformations were significantly longer, 4036.59 ns, 3046.28 ns, and 3382.07 ns, respectively, suggesting that the active conformation (state 4) of EGFR was extremely stable during the simulation, and hardly changed to the inactive states (states 1, 2, and 3).

In the L858R-EGFR state (shown in Figure 5c), the four representative conformations accounted for, respectively, the following: state 1, 0.1%; state 2, 0.1%; state 3, 0.1%; and state 4, 99.7%, with the state 4 conformation differing from both the conventional active and inactive states, while all other states converged towards the inactive conformations converge. The transition times from states 1, 2, and 3 to state 4 were 10.48 ns, 5.16 ns, and 8.61 ns, respectively, indicating that the inactive states (states 1, 2, and 3) can easily transition to the specific conformation (state 4) and the transition rate is fast. This suggests that L858R mutant EGFR has a clear tendency to converge from the inactive state to the specific conformation. In contrast, the retrotransition times from state 4 to the other inactive conformations (states 1, 2, and 3) were 402,737 ns, 3976.64 ns, and 20,901 ns, respectively. These retrotransition times were significantly longer than the transition times from the inactive state to state 4, indicating that the special conformation of state 4 was extremely stable during the simulation and rarely transitioned to the inactive state. Overall, similar to T790M mutant EGFR, the L858R mutation significantly improves the ability of EGFR to converge from the inactive state to a stable conformation (even though state 4 is not typically the active state). This conformational stability and occupancy may provide a structural basis for the enhanced activation function of its mutants, consistent with the notion that mutant EGFR is more readily activated.

In the T790M_L858R-EGFR state (shown in Figure 5d), the four state occupancies were state 1, 7.2%; state 2, 13.7%; state 3, 15.7%; And state 4, 63.4%. States 1, 2, and 4 are conformations that converge to the active state, with state 3 converging to the active state converges to a lesser extent than the other three states. The transition times from states 1, 2, and 3 to the active state 4 were 11.91 ns, 20.19 ns, and 33.81 ns, respectively, which indicated that states 1, 2, and 3 could be converted to the active state 4 relatively quickly, which showed a tendency for double-mutant EGFR to converge to the active conformation more readily than single-mutant or wild-type EGFR.

### 2.3. Interaction Network and Allosteric Pathway Analysis

Although the above MSMs analysis showed that the double mutant (T790M_L858R) EGFR prefers to adopt an active conformation during the simulation, it is unclear how the double mutation transmits allosteric signals from other regions to the αC-helix and A-loop regions to activate EGFR. To further investigate the molecular mechanism caused by EGFR mutation, we used the neural relationship inference (NRI) method to analyze the residue–residue and domain–domain interaction networks of the six regions divided in the mutant (Figure 6a), as well as the specific pathways of allosteric signal transduction.

Figure 6b,c show the residue–residue and domain–domain interaction networks of EGFR in the wild-type and mutant states, represented by the NRI-derived interaction matrix, where the color depth represents the interaction strength (weight). As shown in Figure 6b,c, the interaction strength of the mutant state is higher than that of the wild-type state, especially the double mutant state. Around the A-loop, N-lobe, and ploop regions, this indicates that the mutant significantly enhances the internal interactions of EGFR. The network diagram (Figure 6d,e) more intuitively shows the inferred domain–domain interactions and their directionality. In the wild-type structure, the network is more uniform as a whole, and each domain shows both significant signal-receiving ability (high in-degree) and strong signal-emitting ability (high out-degree). In addition. In contrast, in the mutant structure (Figure 6f,g), there are obvious differences in the interaction strength between the domains, which are manifested in the C-lobe, αC-helix, N-lobe, and loop domains. This indicates that the mutation reorganizes the interaction network and propagates the activation signal through a specific allosteric pathway. By observing the heat map and network diagram, it can be seen that the mutation significantly enhances the interaction strength between the loop and p-loop, A-loop regions, and other domains.

To dissect how oncogenic mutations reshape allosteric communication in EGFR, we applied a shortest-path analysis to the NRI-derived interaction networks, focusing on signal propagation from the A-loop to the P-loop and N-lobe. Each region contains multiple residues, so we selected the three most connected nodes (based on edge weight and centrality) from each domain as signal origins and endpoints. Shortest paths were computed using Dijkstra’s algorithm and visualized via concentric circle diagrams (Figure 7a–d), where all residues are arranged radially by centrality, from the core to the periphery. Arrows represent the shortest directional routes connecting A-loop to its downstream targets. The wild-type network (Figure 7a) exhibited highly centralized, direct paths, indicating compact and efficient interdomain communication. In contrast, the T790M mutant (Figure 7b) showed fragmented signal flow and reduced centrality in key relay nodes, suggesting a loss of network cohesion. The L858R network (Figure 7c) displayed further remodeling, with altered path topology and shifted endpoints, consistent with reorganization of communication routes. The double mutant (Figure 7d) exhibited the most extensive redistribution, with increased path dispersion and emergence of alternative communication hubs, suggesting a synergistic rewiring that enhances network connectivity across domains.

Overall, these results indicate that EGFR mutations not only elongate or shift allosteric signaling paths but also redistribute the network centrality landscape, potentially enabling conformational transitions by relaxing communication constraints. The emergence of alternative signaling routes in mutants may reflect a mechanistic basis for the destabilization of the inactive state and enhanced accessibility to activation-prone conformations.

Following the network-level characterization of signal propagation (Figure 7), we projected the shortest paths onto the three-dimensional structure of EGFR to assess how these communication routes are embedded within the protein’s physical framework (Figure 8). In WT-EGFR, the paths form a structurally coherent arc traversing the N-lobe, closely aligned with buried elements of the kinase core. This suggests a conserved spatial routing mechanism supporting stable intramolecular signaling.

By contrast, in the T790M and L858R mutants, the projected routes deviate from the canonical core and increasingly traverse solvent-exposed or peripheral surfaces. These alterations indicate that the mutations not only disrupt logical connectivity but also reroute physical signal conduction through more flexible or less constrained spatial corridors. The T790M_L858R double mutant exhibits the most pronounced remodeling: communication pathways extend across a widened structural footprint, bypassing original regulatory interfaces and involving alternate surface-exposed residues. These spatial shifts reflect a fundamental reorganization of EGFR’s allosteric wiring—not merely in topological terms but in how signal flows are physically embedded within the structure. By expanding the communication geometry and engaging new residue interfaces, the mutations may facilitate activation by decoupling structural constraints that stabilize the inactive conformation.

## 3. Discussion

Although the structures of wild-type EGFR and its activating mutants have been extensively characterized, the underlying mechanism by which specific point mutations reshape the conformational ensemble and promote sustained activation remains incompletely understood. In this study, we integrated long-timescale metadynamics simulations [14,15], Markov state models (MSMs), and neural relational inference [16,17] (NRI) to systematically dissect how T790M, L858R, and their combination perturb the conformational equilibrium and allosteric communication landscape of EGFR. Our multiscale analysis reveals a dynamic activation mechanism driven by mutation-induced shifts in both kinetic topology and interaction networks.

The MSMs analysis captured a clear redistribution of conformational populations across wild-type and mutant EGFR systems. In the wild-type ensemble, the conformational landscape is dominated by a stable inactive macrostate, with only sparse and transient transitions into active-like conformations. This indicates that, under physiological conditions, EGFR remains kinetically trapped in an autoinhibited state. In contrast, the T790M and L858R mutants, particularly in combination, dramatically reshaped the equilibrium. Active-like macrostates gained substantial occupancy, and transitions toward these states became more frequent and rapid. These observations highlight a dual mechanism of activation: the mutants not only increase the thermodynamic accessibility of active conformations but also reduce kinetic barriers between inactive and active states. Notably, the double mutant exhibited enhanced interconversion between multiple metastable states, suggesting increased dynamic plasticity that further facilitates pathway entry into activation-competent regions. These results delineate how EGFR mutants shift the conformational equilibrium both thermodynamically—by stabilizing active-like macrostates—and kinetically—by accelerating transitions that would otherwise be suppressed in the wild-type ensemble.

While MSMs elucidate the global reshaping of the conformational ensemble and transition kinetics, they do not explain how a single mutation at the ATP-binding cleft propagates its effects through distant regulatory regions. To address this gap, we employed NRI to reconstruct time-resolved interaction networks from the atomic trajectories. This approach revealed that the mutants reorganize the strength and architecture of residue-residue interactions, particularly within key regulatory motifs such as the A-loop, P-loop, and αC-helix. Network centrality analysis identified increased betweenness of several previously peripheral residues, suggesting that the mutations dynamically rewire the underlying allosteric circuit. Crucially, shortest-path analyses revealed that mutations altered not only the topology but also the directional flow of information between distant regions. In wild-type EGFR, communication between the A-loop and N-lobe followed compact, well-centered paths. These signal routes reflected a closed regulatory circuit consistent with stable autoinhibition. In contrast, mutant networks exhibited longer, more distributed paths with increased detours through secondary relay nodes. This dispersal of signal flow weakens the self-inhibitory loop and promotes broader structural flexibility. The double mutant, in particular, showed the emergence of multiple parallel communication channels that bypass the canonical closed conformation bottlenecks. When these paths were mapped onto the three-dimensional structure of EGFR (Figure 8), they aligned with the motion vectors observed in essential dynamics analysis and revealed the spatial footprint of a rewired signaling landscape.

Together, these results support a unified mechanism in which activating EGFR mutations trigger conformational reorganization not solely by destabilizing inhibitory domains, but by actively rewiring the internal communication network to favor activation-compatible dynamics. The T790M and L858R mutations promote a shift from localized to distributed control, enabling information to traverse alternative routes and expanding the conformational repertoire accessible under physiological conditions. The resulting network remodeling not only facilitates domain coupling and motion coherence but also lowers the energetic and topological constraints that otherwise prevent spontaneous activation.

This coupling between conformational redistribution and allosteric network rewiring provides a mechanistic basis for the persistent activation observed in mutant EGFR systems. Notably, resistance-associated mutations may capitalize on latent communication routes to bypass canonical regulatory checkpoints, thereby exploiting hidden allosteric vulnerabilities. From a therapeutic standpoint, strategies that disrupt these dynamically organized signaling architectures—rather than merely occupying static active sites—may offer more durable suppression of oncogenic EGFR activity. This framework highlights the potential of targeting allosteric conduits and interfacial relay hubs as a new strategy to intervene in mutation-driven hyperactivation, with broad implications for the design of next-generation inhibitors across the receptor tyrosine kinase family.

## 4. Materials and Methods

### 4.1. Protein Model Construction

To investigate the structural impact of clinically relevant EGFR mutations, we constructed molecular models based on crystallographic structures of the kinase domain in both inactive (PDB ID: 2GS7 [18]) and active (PDB ID: 2GS2 [19]) conformations. The modeled region spanned residues L703–Q976 (corresponding to L679–Q952 in the original PDB files), encompassing all critical regulatory elements including the αC-helix, A-loop, and P-loop. Oncogenic point mutations—T790M, L858R, and the double mutant T790M_L858R—were introduced into the inactive-state structure (2GS7) using PyMOL [20]. These mutations are among the most frequently observed alterations in non-small cell lung cancer and are known to drive constitutive activation and therapeutic resistance. To restore structural completeness, missing residues were rebuilt using MODELLER-10.4 [21].

All models were subjected to energy minimization using 10,000 steps of the conjugate gradient algorithm under the AMBER99SB-ILDN [22] force field, ensuring the removal of steric clashes and relaxation of local geometries. The minimized structures were then assessed for stereochemical quality and overall structural integrity. These optimized models served as the starting conformations for subsequent metadynamics simulations [23] to probe the mutation-induced conformational transitions.

### 4.2. Metadynamics Simulation

To enhance the sampling of rare but functionally relevant conformational transitions in EGFR, we performed well-tempered metadynamics simulations using GROMACS-2023 [24] patched with PLUMED-2.9.0 [25]. This approach enables exploration of high-energy intermediates and overcomes limitations associated with conventional molecular dynamics.

Each system, including wild-type EGFR and three mutant variants (T790M, L858R, and T790M_L858R), was solvated in a cubic box using explicit TIP3P [26] water molecules. Counterions (Na^+^ and Cl^−^) were added to neutralize the system and maintain a physiological salt concentration of 150 mM. Energy minimization was carried out using the steepest descent algorithm to eliminate steric clashes. Equilibration was performed sequentially under NVT and NPT ensembles, maintaining a temperature of 300 K using a velocity-rescaling thermostat (τ = 0.1 ps) and a pressure of 1 atm using the Parrinello–Rahman barostat (τ = 0.5 ps). All bond lengths were constrained using the LINCS algorithm, and a 2 fs integration timestep was used. Long-range electrostatics were computed with the particle mesh Ewald (PME) [27] method with a real-space cutoff of 1.0 nm. Lennard-Jones interactions were truncated at 1.0 nm, and a long-range dispersion correction was applied to energy and pressure.

Collective variables (CVs) were defined as bias conformational transitions associated with EGFR activation. Specifically, we selected two inter-residue distances: K745 (NZ)–E762 (CD) and K745 (NZ)–D855 (CG), which reflect the spatial coordination between the αC-helix and the activation loop. The difference between these two distances (Δd = d_1_ − d_2_) was used as a single CV to represent the activation-coupled motion. A bias potential was deposited every 500 steps (1 ps) with a Gaussian height of 3.0 kJ/mol, width of 0.15 nm, and a bias factor of 10. A strong harmonic wall (KAPPA = 4000 kJ/mol·nm^−2^) was applied to confine the CV space within −2 to 3 nm, ensuring physical relevance and avoiding oversampling of unphysical states.

Each metadynamics simulation was run for 1 μs to allow extensive sampling of conformational space. By biasing key activation-associated degrees of freedom, this enhanced sampling protocol enabled the system to overcome free-energy barriers and access intermediate or activated-like conformations that are otherwise inaccessible on conventional MD timescales.

### 4.3. Conformational Flexibility and Molecular Motion Analysis

To evaluate the structural stability and dynamic characteristics of EGFR variants, we conducted a series of trajectory-based analyses focusing on conformational deviation, residue-level fluctuations, and collective backbone motions.

Backbone root-mean-square deviation (RMSD) was calculated using the ‘gmx rms’ module [28] in GROMACS, measuring the deviation of atomic positions from the initial minimized structure over time. This metric was used to assess the overall stability of each system throughout the simulation. Additionally, root-mean-square fluctuation (RMSF) values of Cα atoms were computed using ‘gmx rmsf’ [29] to quantify local flexibility across residues, with particular attention to functionally relevant regions such as the αC-helix and activation loop.

To investigate large-scale collective motions, we performed an essential dynamics (ED) analysis via principal component analysis (PCA) [30] of the Cα atom covariance matrix. The covariance matrix was constructed using ‘gmx covar’, and the dominant eigenvectors were obtained and projected with ‘gmx anaeig’. The first few principal components were selected to capture major conformational changes. Porcupine plots were generated to visualize the direction and amplitude of backbone displacements along the first eigenvector, with a focus on regulatory elements including the αC-helix, activation loop, and P-loop.

### 4.4. Markov State Models

To characterize the thermodynamic and kinetic behavior of EGFR activation, we constructed Markov state models [31] (MSMs) based on the metadynamics trajectories of wild-type and mutant systems. MSMs provide a statistical framework that discretizes the high-dimensional conformational space into metastable states and quantifies the probabilities and timescales of transitions among them.

Functionally relevant motions were captured by extracting inter-residue Cα–Cα distances between the αC-helix and activation loop, which served as input features for dimensionality reduction [32]. Time-lagged independent component analysis (TICA) was applied with a lag time of 0.1 ns to isolate the slowest modes of conformational change [33]. The trajectory data were projected onto the top two independent components for subsequent clustering and visualization.

Kinetic discretization was performed by applying the K-means algorithm to the TICA-projected data, generating 1000 microstates that represent distinct conformational configurations. MSMs was then constructed by estimating a transition probability matrix using a lag time of 2 ns, which was selected based on a convergence analysis of implied timescales. Microstates were subsequently coarse-grained into four macrostates using the Perron Cluster Cluster Analysis (PCCA+) method to enable interpretation of large-scale conformational transitions [34].

To ensure the reliability of the model, its Markovianity was validated through the examination of implied timescale convergence and Chapman–Kolmogorov tests [35]. Equilibrium populations of macrostates were derived from the stationary distribution of the transition matrix. Free energy surfaces were reconstructed by projecting the conformational data onto the first two TICA components, allowing the visualization of metastable basins and conformational heterogeneity. In addition, macrostate-level transition networks were constructed based on interconversion probabilities to facilitate the kinetic interpretation of activation pathways.

All MSM-related computations, including dimensionality reduction, clustering, model construction, and validation, were conducted using the PyEMMA-2.5.7 software package [36].

### 4.5. Neural Relational Inference

To analyze mutation-induced changes in residue–residue interaction patterns and allosteric signaling pathways within EGFR, we employed the neural relational inference (NRI) framework—a graph-based deep learning model designed to infer latent interaction graphs from time-series data [37,38]. The method integrates structural dynamics with probabilistic graph inference to capture both spatially proximal and distal dynamic couplings.

Input features were derived from the metadynamics trajectories of each EGFR system. To reduce computational complexity while preserving critical kinetic signals, trajectories were uniformly downsampled to retain 5000 frames, and only Cα atoms of 137 residues were included. Each frame contained 3D Cartesian coordinates and velocity vectors, resulting in a (137 × 6) feature matrix per frame. Features were min–max normalized across the dataset to a [−1, 1] range to ensure consistent scaling. Trajectories were segmented using a sliding window of 50 frames with a 100-frame interval, generating partially overlapping temporal sequences to capture both short- and long-timescale dynamics.

The NRI model comprises a variational encoder and a graph-based decoder. The encoder infers a probabilistic interaction graph among residues, modeling pairwise edges as latent categorical variables. The decoder predicts future states of the system using message passing along the inferred edges. Model training aimed to minimize the reconstruction error between predicted and actual trajectories using a mean squared error (MSE) loss function. Training was conducted using the Adam optimizer with an initial learning rate of 5 × 10^−4^ and batch size of 1. The learning rate was decayed by a factor of 0.2 every 200 epochs over a total of 500 epochs. The model achieving the lowest validation loss was selected for downstream inference. After training, the decoder-derived interaction probabilities were averaged over all segments to obtain an interpretable interaction matrix, which reflects the strength of dynamic coupling between residue pairs.

Interaction matrices were visualized using Cytoscape-3.10.3 [39], and shortest allosteric paths were computed using Dijkstra’s algorithm [40], focusing on signal propagation routes from the A-loop to P-loop and N-lobe regions. Domain-level networks were extracted by grouping residues into predefined structural modules.

## Figures and Tables

**Figure 1 ijms-26-06226-f001:**
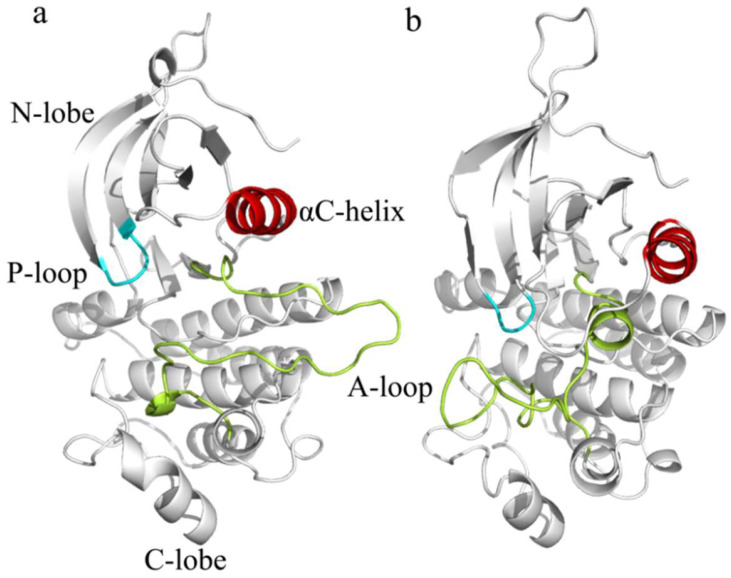
Structural comparison of the epidermal growth factor receptor in its active and inactive states. (**a**) Active conformation of the epidermal growth factor receptor (PDB ID: 2GS2). (**b**) Inactive conformation of the epidermal growth factor receptor (PDB ID: 2GS7). Structures were aligned and visualized using PyMOL-2.6.0. Key dynamic regions, including the αC-helix and the activation loop (A-loop), are color-coded to highlight conformational differences associated with activation. The A-loop is rendered in green, the αC-helix is rendered in red, and the P-loop is rendered in blue.

**Figure 2 ijms-26-06226-f002:**
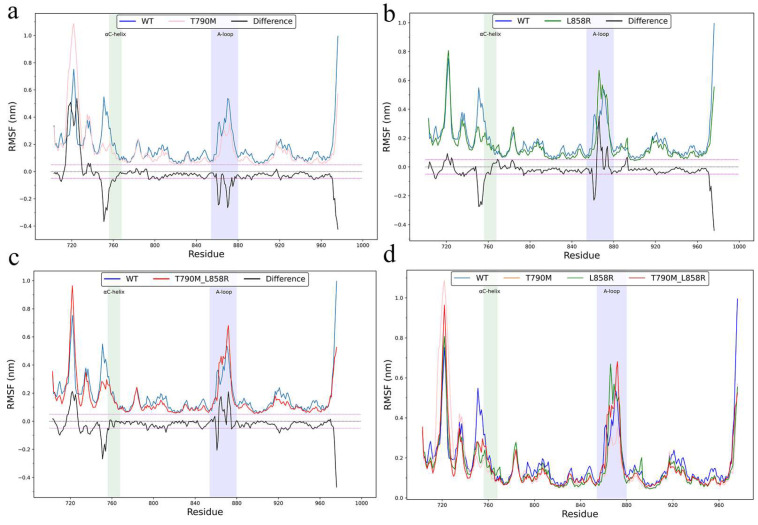
Residue root-mean-square fluctuation (RMSF) curves of epidermal growth factor receptor variants. (**a**) Comparison of RMSF of WT-EGFR with T790M mutant. (**b**) RMSF comparison of WT-EGFR with L858R mutant. (**c**) RMSF comparison between WT-EGFR and T790M_L858R double mutant. (**d**) RMSF profiles of all four systems superimposed: WT (blue), T790M (orange), L858R (green) and T790M_L858R (red). RMSF values were calculated from Cα atoms and plotted as a function of residue number. The black line indicates the difference in RMSF per residue between each mutant and WT-EGFR. Functionally critical regions, such as the αC-helix and the activation loop (A-loop), are marked in green and light blue, respectively.

**Figure 3 ijms-26-06226-f003:**
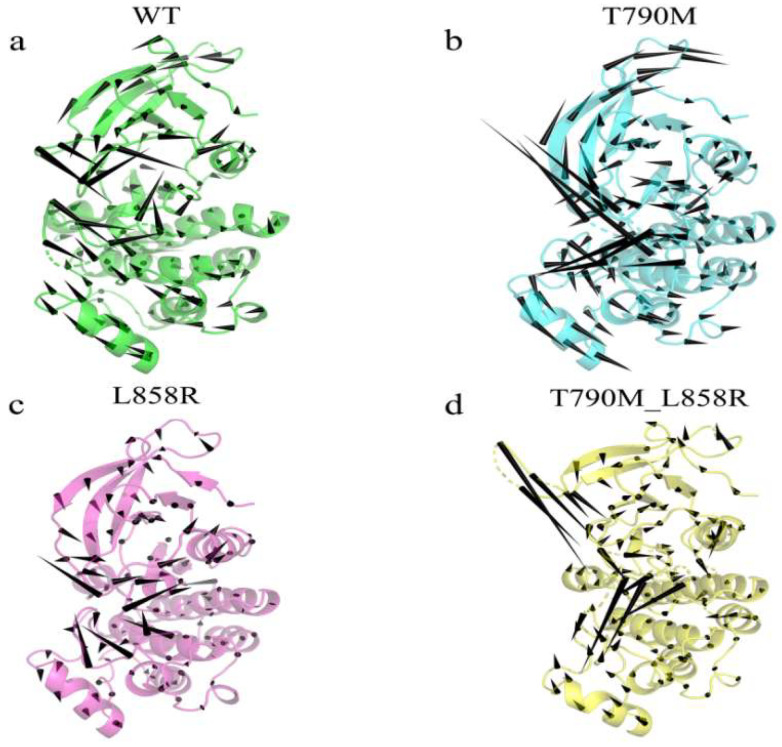
Porcupine plot of WT-EGFR (**a**), T790M-EGFR (**b**), L858R-EGFR (**c**), and T790M_L858R (**d**) produced from the first eigenvector projection. where the cone is plotted on the Cα atom, and its pointing and length ratios indicate the direction of motion and the amplitude of Cα fluctuations, respectively, produced by PyMOL.

**Figure 4 ijms-26-06226-f004:**
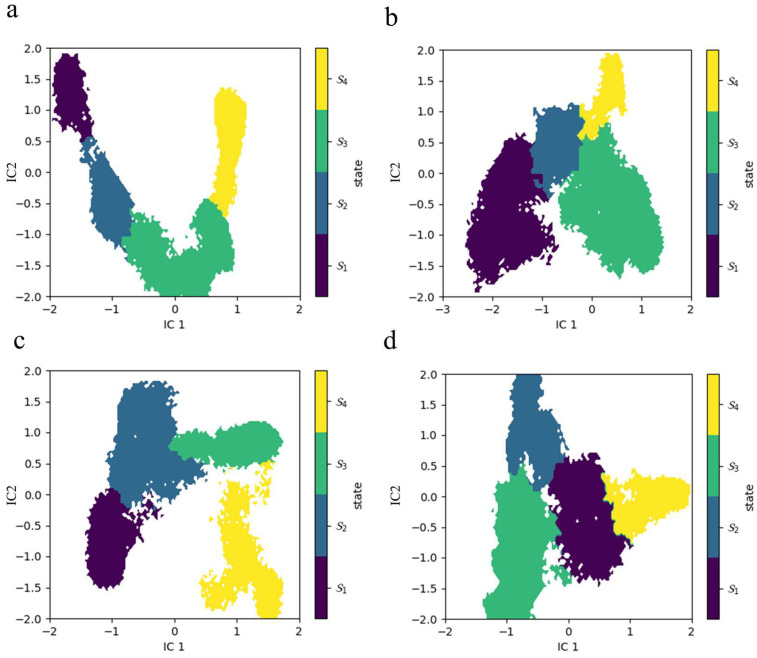
MSMs analysis of EGFR conformational states. (**a**) WT-EGFR, (**b**) T790M mutant, (**c**) L858R mutant, and (**d**) T790M_L858R double mutant. Conformational macrostates identified by the MSMs are color-coded (S1 to S4) and projected onto the first two independent components (IC1 and IC2) from time-lagged independent component analysis (TICA), illustrating the distribution and separation of the different metastable states in the conformational space.

**Figure 5 ijms-26-06226-f005:**
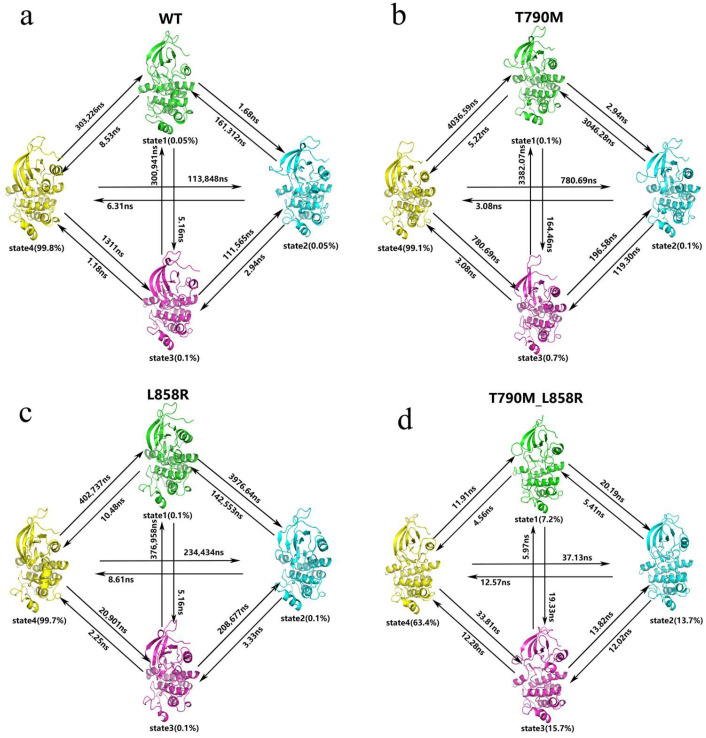
MSMs analysis of EGFR variants. (**a**–**d**) show the macrostate transition networks and representative structures for WT (**a**), T790M (**b**), L858R (**c**), and T790M_L858R (**d**). Each colored node represents one of the four macrostates (S1–S4), annotated with its equilibrium population. Representative conformations are shown adjacent to each state node. Arrows indicate the mean first passage time (MFPT, in nanoseconds) between states, reflecting the kinetic accessibility of interconversion pathways. The asymmetry of transition times reveals directional preferences in conformational transitions. WT-EGFR exhibits a kinetically isolated S4 state, while mutants display enhanced interconversion among active-like states.

**Figure 6 ijms-26-06226-f006:**
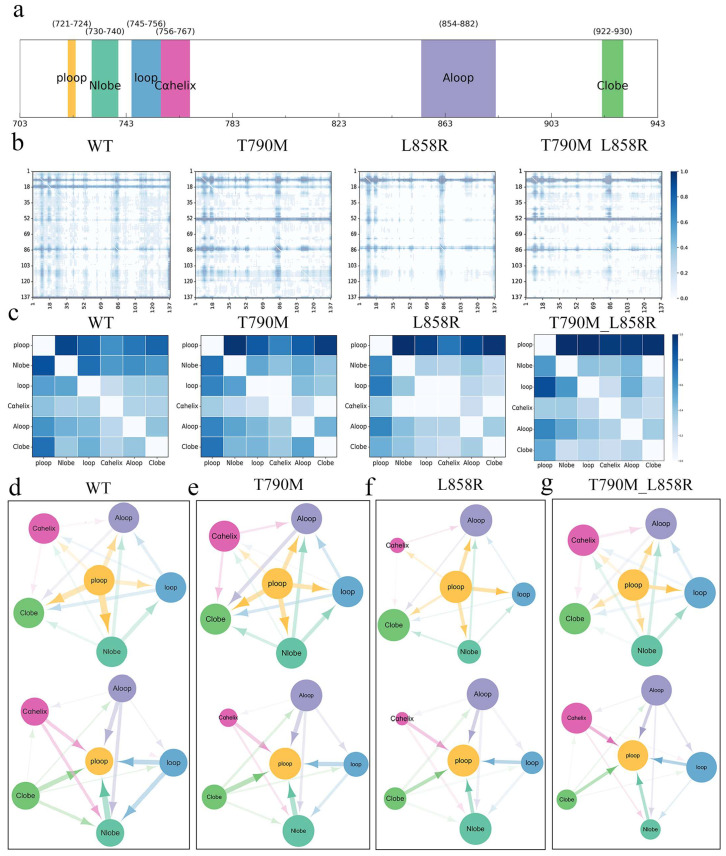
Neural relationship inference (NRI) analysis of EGFR interaction network. (**a**) Domain segmentation of EGFR protein, (P-loop, N-lobe, Loop, αC-helix, A-loop, and C-lobe); (**b**) NRI-derived interaction matrix of EGFR wild-type and mutant, where color intensity represents interaction strength (weight); (**c**) domain–domain interaction matrix of EGFR wild-type and mutant is summarized; (**d**–**g**) network diagrams illustrating the inferred domain–domain interactions and their directionality in (**d**) the WT state, (**e**) the T790M state, (**f**) the L858R state, and (**g**) the T790M_L858R state. The color of the arrowhead corresponds to the color of the starting structure, and the size of the arrowhead indicates the strength of signal transmission from that structural domain to another structural domain.

**Figure 7 ijms-26-06226-f007:**
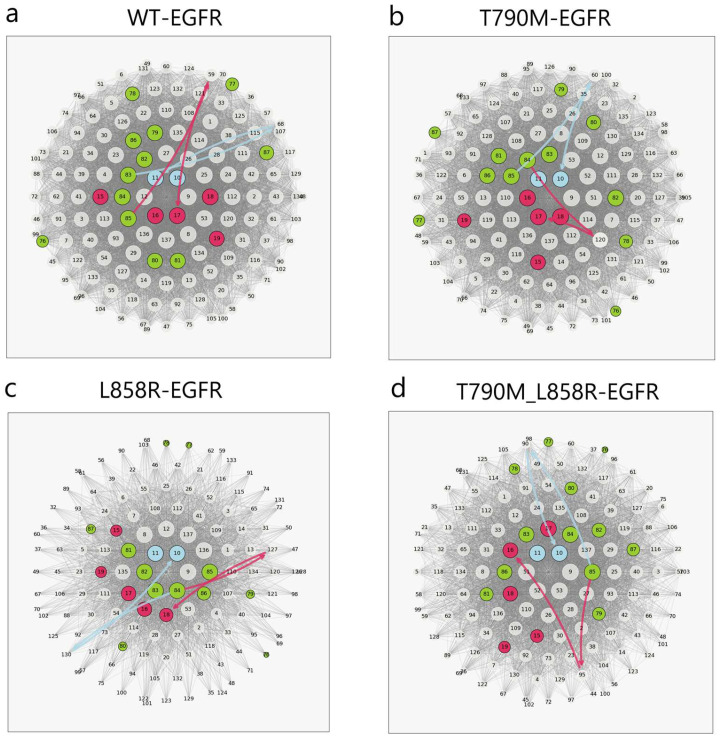
Shortest-path networks of allosteric communication in EGFR. Concentric circle plots display the shortest communication paths from the A-loop to the P-loop and N-lobe, inferred from NRI interaction networks using Dijkstra’s algorithm. All protein residues are shown as nodes, arranged radially by centrality (from center to periphery). Arrows indicate shortest paths, colored by target domain (blue: P-loop; pink: N-lobe; green: A-loop). (**a**) WT-EGFR exhibits short and centralized paths. (**b**) T790M shows dispersed paths and reduced node centrality. (**c**) L858R displays extensive network remodeling. (**d**) The double mutant T790M_L858R exhibits a fully redistributed network, suggesting synergistic rewiring.

**Figure 8 ijms-26-06226-f008:**
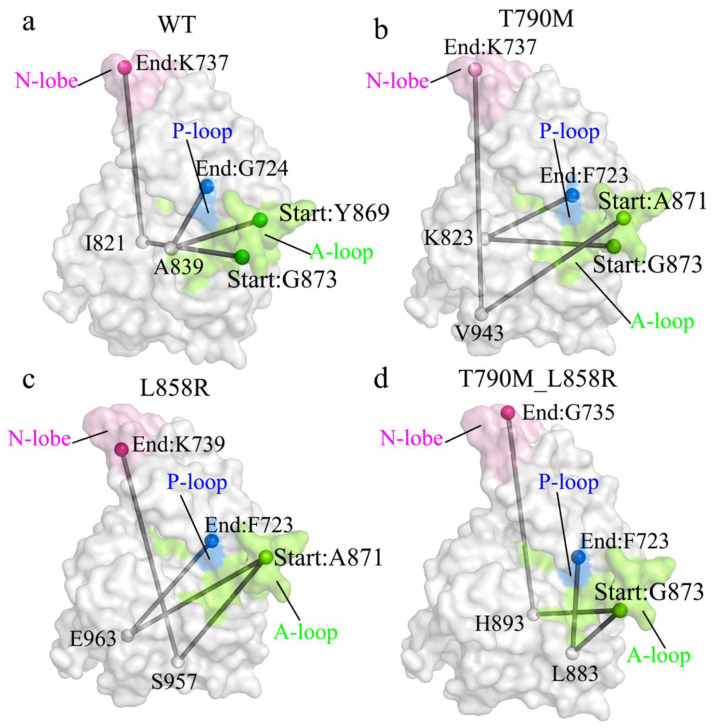
Structural projection of allosteric pathways on the EGFR surface. Representative shortest paths (corresponding to Figure 7) are mapped onto the 3D structures of WT and mutant EGFR. Residues from the A-loop (green), P-loop (blue), and N-lobe (pink) define the source and target regions; gray lines trace the spatial propagation paths. (**a**) WT shows a compact intra-core trajectory. (**b**) T790M mutation displaces the path outward. (**c**) L858R induces pathway redirection and endpoint shift. (**d**) The double mutant exhibits widespread spatial redistribution and path expansion.

## Data Availability

The original contributions presented in this study are included in the article/Appendix A. Further inquiries can be directed to the corresponding authors.

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
