# Peer review of "Conformational Remodeling and Allosteric Regulation Underlying EGFR Mutant-Induced Activation: A Multi-Scale Analysis Using MD, MSMs, and NRI"

_ijms, 2025, doi:10.3390/ijms26136226_

Round 1
Reviewer 1 Report
Comments and Suggestions for Authors
The results reported in this manuscript are a multi-scale structural analysis of EGFR wt and its activation-inducing mutants using MD, MSMs and NRI. It attempts to explain how activating mutations in EGFR deform conformational dynamics and modulate its activity. The authors claim that they provide insights into how these mutations affect the conformational ensemble of EGFR, favoring activation-prone states and facilitating the transition to the active conformation.
They also describe the reorganization of the allosteric network that accompanies these mutations, suggesting new possibilities for therapeutic design. This manuscript presents work at a very sophisticated level of analysis of the dynamic structure of EGFR.
Minor remarks:
Figures 1, 2 and 4 are labeled 1 (a, b) and 2 and 4 (a, b, c, d). Why, if it is not reflected in the description of these figures?
Author Response
Comments :The results reported in this manuscript are a multi-scale structural analysis of EGFR wt and its activation-inducing mutants using MD, MSMs and NRI. It attempts to explain how activating mutations in EGFR deform conformational dynamics and modulate its activity. The authors claim that they provide insights into how these mutations affect the conformational ensemble of EGFR, favoring activation-prone states and facilitating the transition to the active conformation.
They also describe the reorganization of the allosteric network that accompanies these mutations, suggesting new possibilities for therapeutic design. This manuscript presents work at a very sophisticated level of analysis of the dynamic structure of EGFR.
Minor remarks:
Figures 1, 2 and 4 are labeled 1 (a, b) and 2 and 4 (a, b, c, d). Why, if it is not reflected in the description of these figures?
Response:Thank you for your constructive comments on our work and for pointing out its shortcomings. We appreciate you bringing to our attention that the sub-panel descriptions were not reflected in the diagram annotations. We have revised the annotations in Figures 1, 2, and 4 to clearly describe the meaning of each subpanel. For instance, Figure 1 previously failed to clarify whether a and b represented active or inactive structures, respectively. We added information indicating that Figure 1a represents the active structure and Figure 1b represents the inactive structure. Figures 2a–d now show systematic comparisons (WT vs. T790M, WT vs. L858R, etc.). Figure 4 previously lacked four subfigures, which could have misled the reader. We added the missing subfigures and labeled them as follows: Figure 4a represents WT, Figure 4b represents T790M, Figure 4c represents L858R, and Figure 4d represents T790M_L858R. We have also ensured that the graph names are consistent with the text references.
Reviewer 2 Report
Comments and Suggestions for Authors
The current manuscript employs an integrated, multiscale computational framework to dissect how three oncogenic EGFR mutations drive persistent receptor activation by 1 μs well‐tempered metadynamics simulations. By assessing backbone RMSF from simulation trajectories, it shows that the studied mutants increase flexibility of the αC-helix and A-loop, which are the hallmarks of activation‐competent states. With Markov state models (MSMs), it also indicates that the studied mutants cause the shift of macrostate population distribution. Authors also involved neural relational inference (NRI) to reconstruct time-resolved residue–residue and domain–domain interaction networks, demonstrating that mutations rewire allosteric pathways linking the A-loop, P-loop, and N-lobe, both topologically and spatially to relax autoinhibitory constraints and favor activation. The adopted methodology and theoretical consideration are solid and reasonable. There is one minor update required prior to the consideration for final publication. Text explanation for Figure 4 lacks proper indication about the representation of each panel.
Author Response
Comments : The current manuscript employs an integrated, multiscale computational framework to dissect how three oncogenic EGFR mutations drive persistent receptor activation by 1 μs well‐tempered metadynamics simulations. By assessing backbone RMSF from simulation trajectories, it shows that the studied mutants increase flexibility of the αC-helix and A-loop, which are the hallmarks of activation‐competent states. With Markov state models (MSMs), it also indicates that the studied mutants cause the shift of macrostate population distribution. Authors also involved neural relational inference (NRI) to reconstruct time-resolved residue–residue and domain–domain interaction networks, demonstrating that mutations rewire allosteric pathways linking the A-loop, P-loop, and N-lobe, both topologically and spatially to relax autoinhibitory constraints and favor activation. The adopted methodology and theoretical consideration are solid and reasonable. There is one minor update required prior to the consideration for final publication. Text explanation for Figure 4 lacks proper indication about the representation of each panel.
Response : Thank you for your thoughtful and encouraging comments on our manuscript. We appreciate your comments about the lack of clarity in the textual description of the panels in Figure 4. In response, we have revised the legend of Figure 4 to clearly indicate the system represented by each panel:(a) WT-EGFR, (b) T790M mutant, (c) L858R mutant, and (d) T790M_L858R double mutant. We also ensured that references to Figure 4 in the text were consistent with the subfigure numbering.